# Christ Jesus as Object of Cultic Worship in Philippians 3:3b: A Linguistic Study

**Jose Luis Dizon**

Department of Near and Middle Eastern Civilizations, University of Toronto, Toronto, ON M5S 1A1, Canada; luis.dizon@mail.utoronto.ca

**Abstract**

The grammar and syntax of Philippians 3.3 presents a number of ambiguities, particularly in terms of the grammatical object of the verb "worship" (Gk. λατρεύοντες). Most modern translations render the middle phrase of the verse as "worship by the Spirit of God and boast in Christ Jesus" (e.g., CSB, ESV, NLT, NRSV, RSV, etc.). This rendering implies an intransitive use of λατρεύοντες. However, the word order of the verse, as well as the parsing of λατρεύοντες, strongly suggest it is better to understand "Christ Jesus" as the grammatical object of the verb. This essay challenges the prevailing translation of the verse, and argues that the middle phrase should better be translated as "by the Spirit worship and boast in Christ Jesus," to reflect the grammatical relation between "worship" and "Christ Jesus." This re-rendering is highly significant for our understanding of Paul's Christology, as well as contemporary debates over early vs. late high Christology, as it shows points towards Jesus being worshipped as a divine figure by the early Christians, even as early the lifetime of the Apostle.

**Keywords:** Paul; Philippians; Christology; new testament; linguistics; semantics; Bible; Biblical studies

## 1. Introduction

In contemporary Biblical Studies, few disciplines have been as impactful in our understanding the text as Linguistics. Many volumes have been written on the use of various sub-fields of Linguistics (such as Lexical Semantics and Discourse Analysis) to better understand the nuances of the Hebrew and Greek texts of the Bible, and technical commentaries have been produced that aim to examine more closely these Hebrew and Greek texts on a verse-by-verse or clause-by-clause basis. The result is a more precise, technical knowledge of these texts than what had previously been available to classical philologists and grammarians. Despite this heavy use of linguistics to illuminate Biblical texts, a thorough linguistic study of many places in the Greek New Testament (GNT) has yet to be achieved. Such a study, were it to be conducted, would inevitably bring new light to our understanding of the passage under discussion. At the same time, it would challenge our pre-existing translations and exegeses of that passage, and perhaps break the deadlock in some of the longstanding debates on contested issues pertaining to Biblical theology.

One such passage is Philippians 3:3, specifically the middle clause (3:3b) which states οἱ πνεύματι θεοῦ λατρεύοντες καὶ καυχώμενοι ἐν Χριστῷ Ἰησοῦ. This passage presents an important element of Paul's understanding of Jesus, as he is shown to be the one in whom Christians boast. However, this boasting occurs next to a reference to worship (λατρεύοντες), which inevitably leads to the question, "who is the one being worshiped?"

One may be tempted to say that the answer is "Christ Jesus." However, most modern English translations seem to disallow for such an understanding, since they present "worship" as an intransitive verb with no apparent direct object (e.g., "worship by the Spirit of God and boast in Christ Jesus").[1] Furthermore, even the understanding of λατρεύοντες as "worship" is contested, since some English translations choose to translate the verb as "serve" instead, which opens up the possibility that something broader than just worship (in the liturgical sense) is in view.[2] The most common rendering of the clause, though common, obscures a key fact: If one reads the original Greek, the phrase "by the Spirit of God" comes before "worship." Thus, a more exact rendering of the Greek word order yields the following translation: "by the Spirit of God worship and boast in Christ Jesus."[3] Rendered this way, it would appear as if Christ Jesus is not only connected to the verb "to boast," but to the verb "to worship" as well. But can such a conclusion be sustained from a deeper linguistic analysis of the Greek text, or is this merely an optical illusion created by the word order?

The goal of this paper is to argue that modern linguistic study of the GNT can illuminate our understanding of a key theme in Pauline Christology: The worship of Jesus as a divine being. This will be accomplished through a semantic study of λατρεύω as used in Philippians 3:3b, and an analysis of its function within the clause using linguistic analysis, it will be shown that the apostle Paul is speaking of cultic worship of the Church in using λατρεύω, and that the object of this cultic worship is Jesus Christ. This runs counter to the majority of English translations of Philippians 3:3b, which render λατρεύοντες in such a way as to appear to render it intransitive, and would thus necessitate a re-evaluation of the way we translate this verse, as well as how our choice of translation impacts the way the overarching argument of the passage is understood. Finally, this article concludes with a brief discussion of how this re-evaluation also ties into contemporary discussions regarding early Christology.

## 2. Linguistics and Biblical Studies

To begin with, we must distinguish between Linguistics and Philology. To the non-specialist, these terms seem the same, and are often conflated with each other, since both involve the study of language, although it is the former that often gets crowned the distinction of being called the "science" of language (Matthews 2003, p. 1). Though Linguistics and Philology are distinct, they often overlap in the elements of language being studied. It has been said that a clear-cut distinction between the two disciplines "is scarcely possible, since they are parts of a single whole, and each inevitably encroaches upon the other's territory. Both are concerned with speech, and in large part, with the same documents" (Sturtevant and Kent 1928, p. 9).

Nevertheless, we can draw a distinction based on their focii. Philology concerns the "classical" study of languages, and concerns elements such as grammar and syntax, as well as textual and literary criticism. Usually, this takes the form of the study of various written documents. Often, Philologists may take a prescriptive approach to language, which leads to generalizations about "correct" and "incorrect" usages of the language.

By contrast, Linguistics concerns the more "scientific" elements of a language, such as semantics, morphology, orthography, as well as the grouping of related languages (comparative linguistics), as well as the historical development of languages (historical linguistics). Linguists generally take a more descriptive approach to language, observing different usages without making value judgments.

When it comes to the application of Linguistics to Biblical studies, two subfields of Linguistics are especially important, as they provide many key insights that inform modern linguistic studies of Biblical texts: Lexical Semantics and Discourse Analysis. Both

analyze philological insights using linguistics tools. These subfields will be described as follows. A brief discussion of Greek verbal aspect will also be included, as it plays a minor (but relevant) role in examining Greek verbs.

### 2.1. Lexical Semantics

The use of word studies to determine the meanings of a word has been a staple of classical philology long before the advent of Lexical Semantics. For example, the *Theological Dictionary of the New Testament* (TDNT) discusses various Greek words according to their usage in non-Biblical Greek literature, the LXX, and various NT authors. However, James Barr in *The Semantics of Biblical Language* (1961) has pointed out that much of these early linguistic arguments were "unsystematic and haphazard," and that the TDNT authors often fell into the trap of correlating language with mentality—a method of word study that he regarded as "outmoded and scientific," since one cannot neatly align specific word usages with thought structures (cited in Silva 1994, pp. 18–19).[4] Despite the backlash against Barr's work, he has helped bring to light some of the more unhelpful usage of word studies in the early and mid 20th century (Silva 1994, pp. 20–22).

In Lexical Semantics, two insights have helped advance the study of Hebrew and Greek words. First is the concept of the "semantic domain" (or "semantic range"). This can broadly be defined as the full range of possible meanings a given word may have across all the given instances of it in a corpus of writings. Jobes helpfully illustrates this in an appendix to Silva's work, where she points to the GNT's multiple words for worship (προσκυνέω, εὐσεβέω, λατρεύω, σέβομαι, and σεβάζομαι), and shows the similarities and differences in each word's range of meanings, using the different usages of each word (Jobes 1994, pp. 201–11).

Second is a more nuanced understanding of synonymy. Silva notes that there are actually three forms of synonymy: Proper Synonymy, where two words overlap in meaning (e.g., "pretty" and "beautiful"), Improper Synonymy, where two words have contiguous meanings within a broader semantic field (e.g., "move," "walk," and "run"), and Hyponymy, where the meaning of one word is included within another (e.g., "flower" and "rose") (Silva 1994, pp. 119–29). Conversely, these insights also give us a more nuanced understanding of oppositeness, as we may distinguish between *Antonymy*, where two words are semantically related in terms of their opposition to each other (e.g., "short" and "tall"), and *Incompatibility*, where the semantic domain of one word excludes that of others (e.g., "blue," "red," "yellow," etc.) (Silva 1994, pp. 129–32).

These insights into Lexical Semantics will be vitally important in determining what the meaning of λατρεύω is in the context of Philippians 3:3b.

### 2.2. Discourse Analysis

Discourse Analysis is a type of analysis that aims to study texts at a macro-level, going beyond individual phrases and sentences to show how they connect together into a coherent whole. Levinsohn describes Discourse Analysis as "an analysis of language features that draws its explanations, not from within the sentence or word (i.e., the factors involved are not syntactic or morphological), but extrasententially" (Levinsohn 2000, introduction).

Porter notes that the fundamental axiom that drives discourse analysis is that language "is not used in isolated words or even sentences, but occurs in larger units called discourses" (Porter 1999, p. 298). A discourse may vary in size and scope, from a single word, to a letter, to a single book, to a multi-volume work. In discourse analysis of the GNT, a discourse may be a Biblical book, or a corpus of books by the same author. Porter likens constituent elements of a discourse to a pyramid, with the entire discourse at the

top level, and the pericope, sentence, phrase forming descending levels, ending with the individual word at the bottom (Porter 1999, p. 298).

The most significant element of discourse analysis for our present study is the role of word order. Because of the inflected nature of Koine, word order is given less priority than noun cases for establishing the function of words in a sentence. However, this does not mean that word order is irrelevant, as the placement of constituents (subjects, objects and verbs) in a sentence is a vital element of determining what the author is saying, in terms of function as well as the role a given word has in a clause, and consequently in the larger argument of the author.

Porter notes that the GNT displays a number of consistent word order patterns, with some word types often preceding others. While exceptions exist in these patterns, they hold often enough that we may speak of natural word orders in Koine Greek (Porter 1999, pp. 290–92). Porter states:

> On the basis of the statistics and examples cited above, it can be asserted with some plausibility that the Greek of the NT is best described as a linear language, certainly for word order, but also probably for sentence structure. This means that in any given construction the governing (head) or main term has a definite tendency to precede its modifier (Porter 1999, p. 292).

Levinsohn further demonstrates that Koine has a natural word order that it follows visà-vis its constituents. Whereas many contemporary European languages (including English and modern Greek) prefer a Subject-Verb-Object word order (SVO), Koine Greek sentences can more properly be described as naturally preferring a Verb-Subject-Object word order (VSO). Though other word orders are used, Levinsohn states that "*pragmatically*, it is easiest to explain variations in constituent order by taking verb-initial as the default order" (Levinsohn 2000, 2.6)." Later, he discusses how Koine occasionally *preposes* other constituents (i.e., places them earlier in the sentence than their default position) to bring them to focus, and that a verb may even sometimes be placed at the end of a sentence to bring another constituent into focus (or, to bring the verb *itself* into focus) (Levinsohn 2000, 3.6, 3.8.1).

Runge further notes that sentences follow a "natural information flow," wherein discourses tend to move from information that most known to what is least known, while still following the constraints of the language (Runge 2010, 181–89). Hence, objects tend to follow their verbs rather than precede them, since the object is usually the least known datum in a sentence. Following Dutch linguist Simon Dik, however, Runge notes that this natural information flow may be violated under two conditions: (1) to give emphasis to a particular constituent word, or (2) to establish a frame of reference. In such cases, a constituent may be moved in front of a verb even if it violates information flow (Runge 2010, pp. 189–95). Porter gives Romans 7:15–16, 19–20 as an example for how word order may be rearranged to serve an important function in the discourse. There, the relative clauses that serve as the objects for the verbs are placed before the verbs. Thus, ὃ θέλω comes before πράσσω, ὃ μισῶ comes before ποιῶ, ὃ οὐ θέλω comes before ποιῶ, etc. This way, the placement of the object before the verb "gives prominence to the relative clauses, which normally follow their main clauses" (Porter 1999, p. 21) in line with Dik's observations about information flow.

These observations regarding word order and information flow will be vital for determining the grammatical object of the verb in Philippians 3:3b below, as word order yields a vital clue for how the verbs relate to the other constituents in the same clause.

### 2.3. Greek Verbal Aspect

One final insight from modern linguistics that must be briefly mentioned is the increasing recognition that Greek verbs are aspectual rather than tense-based in their usage.

For the longest time, Koine Greek grammarians tended to follow a tense-based theory of understanding Koine verbs, similar to how verb tense functions in English and Modern Greek. However, an increasing number of grammarians have argued that Koine should be understood as following an aspectual understanding of verbs.

One of the earliest major grammarians to argue this was Porter, who defines verbal aspect as "a semantic (meaning) category by which a speaker or writer grammaticalizes (i.e., represents a meaning by choice of a word-form) a perspective on an action by the selection of a particular tense-form in the verbal system" (Porter 1999, p. 21). Later, he argues, "[b]y means of their tense-forms, imperatives and subjunctives… grammaticalize verbal aspect, not temporal reference" (Porter 1999, p. 224). This means that the various Koine tense-forms (present, future, aorist, imperfect, perfect and pluperfect) do not correspond neatly our understanding of verb tenses as indicating discrete points in time (i.e., past, present, or future tense). Instead, the time-value of a verb is determined by its usage within the larger grammatical unit it is contained in, whether sentence, paragraph, or discourse (Porter 1999, p. 21).

Later on, Campbell has argued more forcefully for an aspectual theory of Koine Greek, and the incorporation of aspectual theory in Koine Greek grammar textbooks. In his introduction to aspectual theory, he notes that verbal aspect indicates the viewpoint of the speaker in relation to the action, whether it is viewed from the outside or the inside. He likens perfective aspect to viewing an action from a bird's eye view (e.g., watching a parade from a helicopter in the air), and imperfective aspect to viewing an action from up close (e.g., watching the same parade from the street) (Campbell 2024, pp. 9–11). Further, aspectual theory does not fully do away with the traditional view that tense-forms, particularly in the indicative mood, encode temporal reference (contra a small minority of scholars such as Porter). Rather, the debate is to what extent one can infer time from indicative tense-forms. As this debate is still ongoing, one can find a variety of approaches (Campbell 2024, pp. 26–27).

Although the adoption of this insight into Koine verbs has been slow, it has been gaining ground, with the Greek grammars of Decker (Decker 2014) and the most recent edition of Mounce (Mounce 2019) incorporating verbal aspect. The understanding of how verbal aspect shapes the meaning of a given verb will play a minor but relevant role in our analysis of λατρεύοντες in Philippians 3:3b.

## 3. Philippians 3:3b in Its Context

Having discussed the use of Linguistics in Biblical Studies, we can now prepare to discuss how this discipline can yield insights into the translation and exegesis of Philippians 3:3b. Before this, however, a brief word should first be said about the context of the verse under discussion, as this would help us to frame the parameters of that discussion.

Philippians 3:2–11 is a warning by the Apostle against what he perceives as the threat of the Judaizer heresy. Although this heresy does not loom as large in the epistle to the Philippians as it does in other Pauline epistles (most notably Galatians), we do nevertheless see some concern for combating it. Hence, the warning that we see in verse 2: "Look out for the dogs; look out for the evil workers; look out for the mutilation." The use of the word "mutilation" (κατατομήν) alerts us to the fact that Paul is warning against the Judaizer faction, who are elsewhere called the "the circumcision" (Galatians 2:11–12). The use of κατατομή is an ironic play on the Greek word for "circumcision" (περιτομὴ). George Hunsinger notes that the word κατατομήν is a neologism coined by Paul as a form of sarcasm, and is meant to be act as a pejorative term for the Judaizers (Hunsinger 2020, p. 91). It also recalls Paul's derisive statement about them, that "those who upset you should castrate themselves" (Galatians 5:12).

This brings us to verse 3, where Paul contrasts the false piety of the Judaizers with the true piety of those who adhere to his teaching. He refers to himself and his followers as "the circumcision," thus ironically co-opting the name of his opponents. In using this epithet, Paul is "proclaiming the gospel's superiority over against traditional teachings on the Jewish law" (Cohick 2013, p. 177).

He then describes his followers as being distinguished from the Judaizers in three respects. To quote the original Greek, they are those who "οἱ πνεύματι θεοῦ λατρεύοντες καὶ καυχώμενοι ἐν Χριστῷ Ἰησοῦ καὶ οὐκ ἐν σαρκὶ πεποιθότες." Davis notes that by listing these traits, he is creating a contrast between the opponents, who lack these traits, and himself (as well as the Philippians), who possess them (Davis 1999, pp. 72–73). Bird and Gupta summarize the importance of this trifecta of traits:

> Paul is saying that the sign of belonging to God is not the standard Jewish measures of circumcision, cultus, and confidence in Israel's forthcoming triumph over the pagans. Instead, the currency of covenantal belonging is true obedience (circumcision), being a Spirit-person (worship by the Spirit), and boasting in the Messiah's deeds (no confidence in the flesh) (Bird and Gupta 2020, pp. 120–21).

In considering the meaning of this clause, we may now discuss the two key linguistic questions that concern us in this passage: (1) What is the meaning of the verb λατρεύοντες as Paul uses it in this verse, and (2) what is the grammatical object of the aforementioned verb? We begin our discussion of the first question by considering the role of semantic domains in understanding in what sense λατρεύω is being used here.

## 4. The Semantic Domain of λατρεύω

Of the various verbs for worship in the GNT, λατρεύω is the second most commonly used verb, after προσκυνέω (Jobes 1994, p. 208). There is a wide range of opinions as to what λατρεύω means in the context of the GNT, and Philippians 3:3b in particular. A number of commentators have tried to argue that when Paul uses λατρεύω here, he means "service" in the broad sense of serving God. Joseph Hellerman argues that "serve" is more accurate than "worship" since the latter is too limiting (Hellerman 2015, p. 174). G. Walter Hansen likewise translates the verb as "serve," arguing that it has the connotation of "carrying out of religious duties, especially of a cultic nature" (Hansen 2009, p. 221). Holloway argues that the verb also includes missionary activity, rather than just cultic or liturgical worship (Holloway 2017, p. 154).

Even prior to the advent of Lexical Semantics, it was already understood that the verb λατρεύω had a wide range of meanings. Liddell and Scott's Greek Lexicon, which has been the standard lexicon for classical Greek texts since the mid-19th century, identifies three basic usages of the verb: The first, most basic meaning is "to hire or work for pay; to be in servitude, serve." The second is "to be subjected or enslaved to; serve; obey; to be devoted to." The third, when used in religious contexts, is "to serve the gods with prayers and sacrifices; render due service" (Liddell et al. 1996, p. 1032). For the noun form λατρεία, they similarly note two usages: In secular contexts, it could mean "the state of a hired labourer, service," or "the business or duties of life," whereas in religious contexts, it could mean "service to the gods, divine worship" (Liddell et al. 1996, p. 1032).

By the twentieth century, philological study of λατρεία and λατρεύω has allowed Biblical scholars to identify the different usages of these two words within the context of the Septuagint (LXX) and GNT. Writing for the TDNT, Strathmann notes that, with a few insignificant exceptions, whenever the verb λατρεύω is used in the LXX, it is always used to translate the Hebrew verb עָבַד ("to work, serve") (Clines 1993–2011). The reverse is not true, however, as the TDNT observes that עָבַד is rendered as λατρεύω when the reference is religious, but as δουλεύω when the reference is non-religious (Kittel et al. 1964, p. 60).

He further notes that λατρεύω does not merely refer to worship in the abstract sense, but is used more specifically to refer to cultic, or liturgical, worship (i.e., worship that takes place in the context of a public *cultus*, whether the temple or early Christian house worship). He writes:

> The religious connotation of λατρεύειν is not to be taken, however, merely in a general, abstract, spiritual or ethical sense. It is not enough to say that λατρεύειν has religious significance. One must say that it has sacral significance. λατρεύειν means more precisely to serve or worship cultically, especially by sacrifice. Moses is told that the purpose of the Exodus from Egypt is (Ex. 3:12): λατρεύσετε τῷ θεῷ ἐν τῷ ὄρει τούτῳ, namely, in cultic acts, and especially in sacrifices religious (Kittel et al. 1964, p. 60).

Thus, concludes O'Brien, λατρεύω in the LXX "was used exclusively of religious service—either of the one true God or of pagan deities. This use is determinative of its NT occurrences, where the term has no reference to human relations, much less to secular services" (O'Brien 1991, p. 360).

This meaning then carries over to the GNT. In the majority of cases where it uses the word λατρεύω, the context is worship in the Jerusalem Temple (e.g., Luke 2:47), heavenly worship (e.g., Revelation 22:3), or is a quotation of an LXX passage containing λατρεύω (=עָבַד in the Hebrew). In each of these cases, the meaning of the word is undeniably cultic.

Nevertheless, there are a number of places (particularly in the epistles) where the word λατρεύω is used in a context that is neither temple-focused nor is a quotation of the LXX. One such example is Romans 1:9, where Paul says "God is my witness, whom I worship (λατρεύω) in my spirit in the Gospel of his son, how unceasingly I make mention of you." David Peterson asserts that λατρεύω here refers, not to cultic worship, but to Paul's gospel ministry, which encompasses both preaching and intercessory prayer (cf. Romans 1:11–15) (Beale et al. 2023, p. 873).

Contra Peterson's assertion, the context of Romans 1:9 does not make it unambiguously clear that cultic worship is *not* in view. Although Paul mentions preaching and intercessory prayer, it does not follow that these are the *only* things he meant. Furthermore, preaching and intercessory prayer often occur in the context of liturgical worship, which may explain his use of λατρεύω here. Finally, the context does not support Holloway's suggestion that missionary activity is encompassed by the meaning of the verb. Although missionary work is mentioned elsewhere in the epistle, it is not mentioned in such a way that it is clearly linked to λατρεύω.

In fact, apart from Romans 1:9 and a few other ambiguous instances where cultic worship may or may not be in view, every unambiguous instance of λατρεύω refers to some form of cultic worship, as shown in the instances mentioned above. Viewed in this light, both Romans 1:9 and Philippians 3:3 may be understood to refer to early Christian house worship.

For this reason, Louw and Nida's Greek Lexicon, which groups words based on semantic domains, defines λατρεύω as "to perform religious rites as a part of worship—'to perform religious rites, to worship, to venerate, worship" (Louw and Nida 1996, p. 532). Tellingly, Louw and Nida do not indicate that λατρεύω has a non-ritual usage in any passage of the New Testament. Bird and Gupta concur with this observation. They state: "The word latreuō generally means 'serve,' but its usage in the Greek literature, the LXX, and NT has religious connotations of carrying out religious duties usually of a cultic nature, which renders fitting the translation of 'worship' (ESV, KJV, NRSV, NJV, NASB) (Bird and Gupta 2020, p. 119).

One objection to understanding all these uses of λατρεύω as referring to corporate or liturgical worship is that if that is what Paul and the other New Testament au-

thors intended to convey, they would have used the related word λειτουργέω instead of λατρεύω. One might expect λειτουργέω to be more specifically religious in connotation than λατρεύω. However, Louw and Nida note that this is not the case, stating that "In the NT λειτουργέω and λειτουργίας (53.13) are less specifically religious in connotation than λατρεύω and λατρεία (53.14)" (Louw and Nida 1996, p. 532). In fact, of the three instances of λειτουργέω in the NT, two are used in the context of corporate or liturgical worship (Acts 13:2, Hebrews 10:11), whereas one is used in the generic sense of serving other people (Romans 15:27). The same is true of the nouns λειτουργός and λειτουργία, which are used twice in Philippians in the generic sense of "fellow-worker" (2:25) and "service" (2:30), respectively. Based on this usage, we cannot definitively say that the New Testament authors would have used λειτουργέω instead of λατρεύω if their intention was to convey worship.

We must also consider the relationship between λατρεύω and the most common word for "worship" in the LXX and GNT, which is προσκυνέω (noun form: προσκυνησής). This verb, according to Jobes, has the widest semantic range of all the Koine Greek verbs for "worship" (Jobes 1994, p. 205). Both προσκυνέω and λατρεύω are often translated as "worship" in English, though in some instances, προσκυνέω can be used to refer to non-religious obeisance as well, as in Genesis 23:12 (LXX) where Abraham "bowed (προσεκύνησεν) to the people of the land." Jobes notes that προσκυνέω may be used in one of three senses in the GNT: (1) worship of divinity, (2) obeisance to a political ruler, and (3) making of an entreaty or request (Jobes 1994, p. 205). Adding to this, Lozano also notes that it could be used as a simple respectful greeting of an elder (Lozano 2019, p. 1).

While in most cases, it is easy to tell which sense is meant, some cases are not so clear cut. This is especially the case when the verb is applied to Jesus, of which Lozano identifies 15–16 instances (Lozano 2019, p. 2). For example, when the Magi bow down (προσεκύνησαν) to Jesus in Matthew 2:11, Jobes posit that this is an example of political obeisance (Jobes 1994, p. 206), and Lozano concurs, noting the contrast with Herod, the arrival of royal dignitaries, and the giving of luxury gifts suited for kings as evidence (Lozano 2019, pp. 53–54). By contrast, Greeven argues that it refers to worship of deity, stating: "The proskynesis of the wise men (Mt. 2:2, 11, assumed in 2:8) is truly offered to the Ruler of the world" (Kittel et al. 1964, p. 764). Louw and Nida agree with this assessment and state that in the context of Matthew 2, "to express by attitude and possibly by position one's allegiance to and regard for deity—'to prostrate oneself in worship, to bow down and worship, to worship" (Louw and Nida 1996, p. 539). In other cases, Jesus does receive προσκυνησής in the form of divine worship, but in such contexts, Jobes argues, "it is that supernatural response, and not the word προσκυνέω, that contributes the sense that Jesus is worthy of worship" (Jobes 1994, p. 205). Building upon this, Lozano notes that in Matthew, Luke, and John, when the disciples offer proskynesis to Jesus, it is in a context where Jesus assumes divine prerogatives and powers, lending evidence to a divine worship interpretation of προσκυνέω (Lozano 2019, pp. 81–82, 99, 115–16, 169–71).

Moreover, προσκυνέω and λατρεύω display some level of improper synonymy, as the two verbs sometimes occur together, as seen in Matthew 4:10: "You shall worship (προσκυνήσεις) the Lord, your God, and him alone shall you serve (λατρεύσεις)." In such cases, the former verb denotes an action that takes place within the context of the latter verb (i.e., cultic worship involving prostration). However, it should also be noted that none of the instances of προσκυνέω as applied to Jesus in the Gospels are paired with λατρεύω in this way. This does not mean that προσκυνέω does not denote worship in such cases, only that such instances of worship do not occur in a cultic context, such as the Jerusalem Temple, and in the two instances in the NT where λατρεύω is used in such a

way that Jesus may be construed (albeit ambiguously) as the object of the verb (Philippians 3:3 and Revelation 22:3), the verb προσκυνέω does not occur.

Finally, it is instructive that, whenever λατρεύω is used with an identifiable direct object, it is either the true God or false idols—both of which can be classed as religious worship. This is contrasted with προσκυνέω, whose object is not always religious in character. Nowhere in the NT is λατρεύω used to denote service to a non-religious object, such as an ordinary human being (unless said human being is being made into an idol). This is crucial because it informs us that, whatever the grammatical object of λατρεύοντες in Philippians 3:3b may be, that object must be understood to be divine from the perspective of those rendering worship.

In summary, the verb λατρεύω has a narrower semantic range than προσκυνέω. Although it may denote service to other human beings in secular Greek literature, in virtually all unambiguous cases in the LXX and GNT, it denotes cultic worship rendered to a deity and should thus be understood as such in Philippians 3:3b.

## 5. The Object of λατρεύοντες in Philippians 3:3b

Having established the meaning of the word λατρεύω, we may now turn our attention to its usage in Philippians 3:3b, where it appears in substantival participle form λατρεύοντες. In order to determine the meaning of the clause as a whole, one must ask whether λατρεύοντες is being used intransitively or transitively, and if the latter, what the grammatical object of the participle is.

In turning to the answer, there are four options that need to be considered: (1) that θεοῦ functions as the grammatical object of λατρεύοντες, (2) that πνεύματι θεοῦ functions as the grammatical object of λατρεύοντες, (3) that λατρεύοντες is intransitive (i.e., has no grammatical object), and (4) Χριστῷ Ἰησοῦ is the grammatical object of λατρεύοντες. We will examine each option in turn to see how plausible it is in light of our current linguistic knowledge.

### 5.1. Option 1: θεοῦ as the Object of λατρεύοντες

The first option to consider is that the word "God" (Gk. Θεοῦ) serves as the direct object of λατρεύοντες. This is the preferred view of Hawthorne and Martin, who translate the phrase as "worship God by his Spirit" (Hawthorne and Martin 2004, p. 175). The main problem this option runs into is that Θεοῦ in Philippians 3:3b is in the genitive case, whereas λατρεύω requires its object to be in the dative case. This can be clearly seen in all the verses in the New Testament where λατρεύω has an identifiable direct object (e.g., Acts 7:7, 42, 24:14, Hebrews 13:10, Revelation 22:3, etc.).

To counteract this problem, some later manuscripts change the word for "God" into the dative case (θεω), the most significant being a later corrector to Codex Sinaiticus (ca. mid-4th century CE) (Nestle et al. 2012, p. 608). This is reflected in both the *Textus Receptus* as well as the Latin Vulgate (which reads "spiritu Deo servimus").[5] Thus, English translations based on either the Textus Receptus or the Vulgate (such as the DRB, KJV and NKJV) read "worship God by his Spirit," or some variation thereof. A lesser variant can be found in manuscript P46, which removes the word for "God" altogether and simply reads as πνεύματι λατρεύοντες. According to Metzger, this "is to be explained as due to accidental oversight" (Metzger 1994, p. 547). As for the reading πνεύματι θεω λατρεύοντες, he states that this variant was introduced as a scribal emendation in order to supply an object for the verb, as well as to bring the reading in line with Romans 1:9 and 2 Timothy 1:3, both of which have God as the object of λατρεύω (Metzger 1994, p. 547). In addition, the Byzantine Majority text agrees with the NA28, SBLGNT, THGNT and UBS5 in reading

πνεύματι θεοῦ (Pierpont et al. 1995). Thus, all the textual evidence militates against the use of the dative θεω as the original reading in Philippians 3:3b.

Having ruled out the textual variant, and with the genitive Θεοῦ being almost certainly the original reading, the grammatical evidence weighs heavily against "God" as the object of λατρεύοντες on grammatical grounds.

### 5.2. Option 2: πνεύματι θεοῦ as the Object of λατρεύοντες

One alternative option that may be considered is that "the Spirit of God" (πνεύματι θεοῦ) is the object of λατρεύοντες. Since πνεύματι is in the dative case (as λατρεύω requires), it avoids the grammatical problem posed by the first option above. In fact, Hellerman suggests that the reading Θεω (which was discussed earlier) arose to avoid the possibility that readers might assume that the Spirit of God is the object of worship in the verse (Hellerman 2015, p. 173). Bockmuehl also briefly suggests this as a possible interpretation, before dismissing it on the grounds that such a usage is "unparalleled" (Bockmuehl 1997, p. 192).

Although positing πνεύματι θεοῦ as the object of λατρεύοντες is grammatically possible, it does run into a different type of difficulty, this time syntactical: In the clause under discussion, πνεύματι θεοῦ comes before λατρεύοντες, rather than after it. While it is possible for the object of a verb to precede it in a sentence in Koine Greek (e.g., Exodus 4:23 LXX, where the object pronoun μοι comes before λατρεύσῃ), such constructions are generally rare and generally not expected, as it would violate the natural information flow of the clause. It also violates the aforementioned preference in Koine sentences to place the verb first in a sentence. For the clause in question to follow a SOV word order, it must be demonstrated that the preverbal placement of the object is done either for emphasis or to establish a frame of reference (Runge 2010) or to bring the object into focus (Levinsohn 2000). In the case of πνεύματι θεοῦ, it seems to have been placed before the verb, not because it is the object, but rather to establish the Spirit as the instrument or sphere by which worship and boasting take place (more on that below).

It should also be noted that nowhere in Paul's epistles (or in the New Testament more generally) is the Spirit ever presented as the object of worship. When the Spirit is mentioned in conjunction with worship, it is usually to indicate that the Spirit serves an instrumental role in the worship of God. We see this in John 4:24, where Jesus says that the true worshipper of God "must worship in Spirit and truth" (ἐν πνεύματι καὶ ἀληθείᾳ δεῖ προσκυνεῖν). Another example is Romans 1:9, where Paul says "God is my witness, whom I worship in my Spirit" (ᾧ λατρεύω ἐν τῷ πνεύματί μου). In both cases, worship occurs in/through the Spirit, but the Spirit is not the recipient of the worship.

So what is the purpose of the dative case for πνεύματι in Philippians 3:3b, if it is not as an object for λατρεύοντες? According to Hellerman, the use of the dative for "the Spirit of God" can be understood in two senses: Either as a (1) dative of sphere "in the sphere of, dominated by, the Spirit," or (2) as a dative of instrumentality "by the Spirit of God." Of these two, he argues that the latter is more likely, although this is based on his assumption that λατρεύοντες is to be understood as "serve" (Hellerman 2015, p. 173). Bockmuehl likewise endorses the instrumental dative (Bockmuehl 1997, p. 192). Which sense the dative is being used does not ultimately change the meaning of the verse, and one could plausibly argue that all three senses are being meant all at once, as Bird and Gupta suggest (Bird and Gupta 2020, p. 119).

Finally, it should be noted that of all the commentaries on Philippians surveyed over the course of studying this issue, not a single commentator has suggested that πνεύματι θεοῦ is the object of λατρεύοντες. While lack of supporters does not automatically mean that a certain proposition is false (after all, this very paper is itself arguing for a novel

interpretation), this factor, when combined with all the other factors that have just been discussed above, produces a decisive case against taking this second option.

*5.3. Option 3: λατρεύοντες as an Intransitive Verb*

On the surface, the suggestion that λατρεύοντες has no grammatical object in Philippians 3:3b seems eminently plausible. After all, of the 21 usages of λατρεύω in the GNT, no less than four instances are unambiguously intransitive (Luke 2:37, Acts 26:7, Hebrews 9:9, and 10:2), which shows that there is precedent for an intransitive use of this verb. Granted, this does *not* mean there is no *implied* object (after all, worship must be directed towards someone), but that no grammatical object is explicitly stated—either because the object of worship is obvious or is not the focus of the author (or both).

We have also seen how most English translations (with a few exceptions) render 3:3b in a manner that appears to make interpreting "worship" as an intransitive verb the most natural conclusion. However, this appears to be less the result of a conscious decision to interpret the verb as intransitive and more of a desire to render the Greek text in a manner that flows more idiomatically in English, since it would appear syntactically awkward to place the adverbial phrase "by the Spirit of God" before "worship" in English, as opposed to after it.

Furthermore, of all the options that are being put forward, this option has the greatest number of commentators and scholars in its favour. In fact, whenever the question of what the grammatical object of λατρεύοντες is raised in a given commentary, the answer given is almost always that it is being used intransitively. Most of the time, this position is stated matter-of-factly, without any supporting argumentation. The notable exception to this rule is Hellerman, who argues that the participle is intransitive because the object of worship (or service) is not the issue, but rather, the manner of worship, since both Paul and his Judaizer opponents worshipped the same God. He even goes so far as to argue that to try and supply an object is to obscure what Paul's true emphasis in this verse is ([Hellerman 2015](), p. 173). Even commentators that do not explicitly advocate for this option nevertheless imply or assume it, as seen numerous commentaries (see e.g., [Bockmuehl 1997](), p. 192; [Fee 1995](), pp. 288, 298; [O'Brien 1991](), pp. 360–61, among others).

However, Hellerman's argument appears to be a *non sequitur*, since it is not entirely clear why supplying an object to λατρεύοντες would obscure the point Paul is trying to make here. While it is true that Paul's point is that true believers worship by the Spirit of God (as opposed to the Judaizers who rely on the Law and their own righteousness), that need not rule out other points of emphasis. Perhaps in supplying an object to λατρεύοντες, Paul also intends to emphasize *whom* true worship is directed towards.

Also, while it is true λατρεύω may be used intransitively, it does not follow that it is being used that way in this verse. One of the differences between Philippians 3:3b and these other instances is that in those other instances, λατρεύω is *clearly* intransitive (i.e., there are no candidates for direct object within the verse). That is not the case for the verse under consideration, wherein we do have at least one potential candidate for direct object. In addition, it should be noted that the other two participial clauses in the verse have explicit direct objects. If λατρεύοντες was the only intransitive participle in the verse, that would create a potential asymmetry in Paul's argument.

Furthermore, the theory that λατρεύοντες is being used intransitively here does not consider that there is, in fact, a grammatically plausible object for the participle within the clause—one which is in the dative case—and that is Χριστῷ Ἰησοῦ. Furthermore, its placement after the verb in the clause means that it follows the natural word order of Greek as posited by Levinsohn, as well as the natural information flow of the clause as discussed

by Runge. While this alone is not definitive, in the context of Philippians 3:3b, it does create room for a case to understand Χριστῷ Ἰησοῦ as the object.

To make the case for the intransitive, one would have to ask whether there are any reasons why Χριστῷ Ἰησοῦ could *not* be the object, which is the question we will now proceed to ask.

### 5.4. Option 4: Χριστῷ Ἰησοῦ as the Object of λατρεύοντες

The suggestion that Paul intended to convey that Χριστῷ Ἰησοῦ is the object of λατρεύοντες has not heretofore been seriously considered by commentators of Philippians 3:3. In my survey of commentators, only Witherington translates 3:3b in such a way as to make Χριστῷ Ἰησοῦ the object of λατρεύοντες, and even then, he does not provide an exegetical argument (Witherington 2011, p. 185).[6] Most others assume that Χριστῷ Ἰησοῦ is only the object (direct or indirect) of the participle καυχώμενοι.[7] However, there are a number of factors that would lead us to believe that in the context of 3:3b, Χριστῷ Ἰησοῦ is actually functioning as the grammatical object of *both* verbs.

To begin with, it should be noted that both λατρεύω and καυχάομαι require their direct object to take the dative case, which allows the two verbs to share the same direct object within the same clause without having to repeat the object in two different cases.[8] Furthermore, λατρεύοντες and καυχώμενοι are parsed nearly identically: Both are present substantival participles in the nominative masculine plural. The significance of this, according to Campbell, is that although participles function as verbal nouns, there is verbal nuance to them when used substantivally (Campbell 2024, pp. 152–56). The only difference in parsing is that λατρεύοντες is in the active mood, whereas καυχώμενοι is in the middle mood, although as a deponent, it is functionally active, so the difference disappears in English translations.[9]

Another important note is that both verbs are connected by καί. This may seem like a rather trivial observation, but it matters a great deal from the perspective of discourse analysis. Runge notes that the connector καὶ, although often translated into English as "and," the word καὶ is used to associate words with each other. It "connects two items of equal status, constraining them to be closely related to one another" (Runge 2010, p. 24). This is especially the case in discourses where asyndeton (the linking of related clauses without the use of connectives) is mainly used, since then the clauses joined by καὶ become more closely bound together than the clauses that lack καὶ. As Levinsohn notes, in these cases, καὶ "constrains the material it introduces to be processed as being added to and associated with previous material" (Levinsohn 2000, 7.3).

As a result, the connection of two participles with καὶ indicates a compound unity. We see such a case in passages about table fellowship, where the phrase "eating and drinking" (Gk. ἐσθίων καὶ πίνων) occurs, such as in Matthew 11:29. Although the two participles are conceptually distinct, they are parsed identically and connected by καί to show that they form a larger compound action. Another example of this is Ephesians 5:19 where Paul says λαλοῦντες … ἄδοντες καὶ ψάλλοντες τῇ καρδίᾳ ὑμῶν τῷ κυρίῳ. Here, λαλοῦντες, ἄδοντες, and ψάλλοντες are all used to refer to the singing of praises to the Lord, with each participle being semantically distinct yet being imperfectly synonymous with one another.

An even more apropos example of two participles in a compound unity (which also share a common object) can be found in the Byzantine version of Luke 24:53. Whereas the NA28 simply reads εὐλογοῦντες τὸν θεόν, the Byzantine text has the longer reading αἰνοῦντες καὶ εὐλογοῦντες τὸν θεόν (Pierpont et al. 1995). Metzger explains that this is due to the conflation of two sets of manuscripts: one set that reads εὐλογοῦντες, and another that reads αἰνοῦντες (Metzger 1994, pp. 163–64). In this conflated reading, we

have two participles with the same parsing (present active masculine nominative plural), indicating that they are parts of the same compound action. Furthermore, both share the same direct object: τὸν θεόν. Here we see a direct parallel with Philippians 3:3b, where a single object governs two coordinated participles, both having to do with divine worship, in the context of a shared argument structure.

We also see a very similar parallel example in Colossians 1:10, where Paul encourages his readers to "in every good work bear fruit (καρποφοροῦντες) and increase in (αὐξανόμενοι) the knowledge of God." Here, both καρποφοροῦντες and αὐξανόμενοι have "the knowledge of God" (τῇ ἐπιγνώσει τοῦ θεοῦ) as their object, with "in every good fruit" (ἐν παντὶ ἔργῳ ἀγαθῷ) acting as an adverbial modifier to καρποφοροῦντες. This verse is perhaps the closest parallel to Philippians 3:3, and furnishes solid evidence of two participles sharing a common grammatical direct object.

In fact, the connection between λατρεύοντες and καυχώμενοι is strengthened by one other fact: The phrase οἱ πνεύματι θεοῦ λατρεύοντες καὶ καυχώμενοι fulfils all the requirements of Sharp's Rule. This rule, named after Granville Sharp (who first observed it around the late eighteenth century), states:

> When the copulative καὶ connects two nouns of the same case, [viz. nouns (either substantive or adjective, or participles) of personal description, respecting office, dignity, affinity, or connexion, and attributes, properties, or qualities, good or ill], if the article ὁ, or any of its cases, precedes the first of the said nouns or participles, and is not repeated before the second noun or participle, the latter always relates to the same person that is expressed or described by the first noun or participle: i.e., it denotes a farther description of the first-named person…. (quoted in Wallace 1996, p. 271)

In other words, the two nouns contained within Sharp's identified construction are considered to be one and the same, for all intents and purposes.

In his comprehensive study of Greek grammar and syntax, Daniel Wallace identifies Philippians 3:3b as an example of this type of construction, which he calls an "article-substantive-καί-substantive" (TSKS) construction (Wallace 1996, p. 283). This may seem counter-intuitive since λατρεύοντες and καυχώμενοι are participles rather than nouns *per se*. However, these two substantival participles function as nouns in the verse. In fact, earlier in his grammar, Wallace identifies multiple TSKS constructions where adjectives or participles are functioning as substantives. Although the rule was originally formulated for personal substantives, he notes that participial substantives that refer to persons can behave in such a way as to conform to the rule, as long as the syntactical structure is parallel and no second article is used (Wallace 1996, pp. 274–75). Furthermore, Wallace notes that the presence of other elements within the TSKS construction (in this case, the adverbial phrase πνεύματι θεοῦ) does not negate the rule (Wallace 1996, p. 275).

The identification of οἱ πνεύματι θεοῦ λατρεύοντες καὶ καυχώμενοι as fulfilling Sharp's Rule is highly significant. Although this does not mean worshipping and boasting are the same action, it does mean that those who are worshipping and boasting are one and the same group. Furthermore, the close connection between the two participles indicates that the group that is worshipping and boasting does both simultaneously, and if both actions are happening simultaneously, then it becomes eminently plausible to see "Christ Jesus" as the object of both actions.

We can thus see the overall flow of Philippians 3:3 much better if we place it in the form of the following sentence diagram:

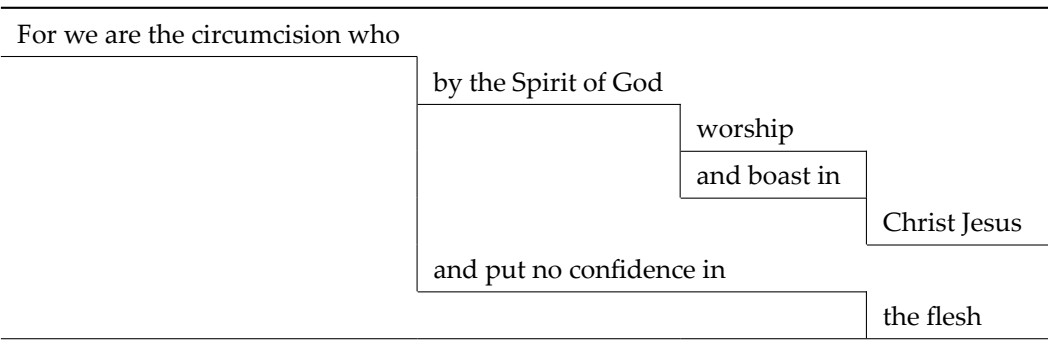

Typically, this clause would be broken up into three participial phrases, each of which is separated by καί: (1) οἱ πνεύματι θεοῦ λατρεύοντες, (2) καυχώμενοι ἐν Χριστῷ Ἰησοῦ, and (3) οὐκ ἐν σαρκὶ πεποιθότες (e.g., O'Brien 1991, pp. 360–64). However, if the grammatical argument for seeing Χριστῷ Ἰησοῦ as the shared object of the two preceding participles holds, then it would be better to treat the first two phrases as a single phrase, which describes what true Christ followers do in two parts. The third phrase would remain separate by virtue of having a different object (σαρκὶ), and functions as an antithetical parallelism to the preceding phrase (i.e., this is what true followers do not do). For Paul, to put confidence in the flesh is to negate true worship and boasting in Christ, which is why the two sets of actions are contrasted with one another.[10]

By diagramming the verse this way, we can clearly see the flow of Paul's thought and argumentation, and how he presents us with two verbs, one object: Christ Jesus, both worshipped and boasted in by "the circumcision."

## 6. Conclusions: Implications for Pauline Christology

The use of modern linguistics to analyze the various syntactical features of the GNT—particularly from the lens of lexical semantics and discourse analysis—has the potential to shed new light on the translation and exegesis of many passages of the NT, and shed light on implications of those passages for ongoing discussions regarding the various themes of the NT. One such passage is Philippians 3:3b. We have seen how linguistically, the participle λατρεύω can be understood in all of its various uses in the NT as being connected in some way to cultic or liturgical worship. Furthermore, the structure of Philippians 3:3b lends itself to understanding Χριστῷ Ἰησοῦ as the grammatical direct object of λατρεύοντες. In other words, Paul is saying that Christians, being the true circumcision, offer cultic worship to and boast in Christ Jesus.

This understanding invites a re-evaluation of our translations of Philippians 3:3. Most contemporary English translations attempt to smoothen the word order of the text by rendering it as "worship by the Spirit of God and boast in Christ Jesus." However, such a rendering, by altering the word order, obscures the natural information flow of the passage, and thus obscures the intended object of the verb "worship." A corrective to this would be to follow the original Greek word order more closely, rendering clause b as "by the Spirit of God worship and boast in Christ Jesus." While this has the disadvantage of sounding less natural in Greek, by preserving the original word order, it also preserves the intended message of the whole verse. Alternatively, by placing the adverbial modifier at the end of the clause, one could create a more idiomatic translation while still presenting "Christ Jesus" as the object of both verbs. Such a translation would then read, "worship and boast in Christ Jesus by the spirit of God."

This understanding is not merely linguistic, but also has relevance to the ongoing discussions regarding NT Christology, and particularly the debate over early versus late high Christology. If Paul is understood to be advocating for the cultic worship of Jesus

in Philippians 3:3b, then this runs counter to the suggestion of such figures as James D.G. Dunn, who state that cultic worship was only ever given to God the Father, and never to Jesus (Dunn 2010, pp. 13–14). It does, however, support the contention of early high Christology advocates such as Richard Bauckham and Larry Hurtado, who state that the earliest Christians (including Paul) worshipped Jesus in a manner that had previously been reserved to God alone (Hurtado 2005, p. 53; Fletcher-Louis 2019, p. 4).

Thus, the use of linguistics in assessing the meaning of Philippians 3:3b provides us with valuable insight into Pauline Christology, and early Christology more broadly. It points us in the direction of a high Pauline Christology, where Jesus can be the recipient of the same class of worship that the ancient Jews offered to Yahweh in the temple cultus, only this time it takes place among early Christians in the context of their own liturgical worship practices.

This study will hopefully spur on similar linguistic studies of other passages that may potentially augment our understanding of New Testament Christology, early Christian worship practices, and the interplay between the two.

**Funding:** This research received no external funding.

**Institutional Review Board Statement:** Not applicable.

**Informed Consent Statement:** Not applicable.

**Data Availability Statement:** Not applicable.

**Conflicts of Interest:** The author declares no conflicts of interest.

## Notes

[1] This translation or some variation of it is used in the ASV, CSB, ESV, NABRE, NASB, NET, NRSV, RNJB, and RSV, among others. Some notable exceptions to this are the DRB, KJV, NKJV, for reasons which will be discussed below. For a comparison of the different translations, see "Philippians 3:3," (Bible Hub 2025), https://biblehub.com/philippians/3-3.htm (accessed 24 June 2025).

[2] E.g., CPDV, GWT, HCSB, NIV, etc.

[3] The reason the verbs have been translated this way is to fully capture their status as substantival participles, which will be explained later.

[4] Despite Barr and Silva's criticisms, it is this author's opinion that the TDNT may still be employed fruitfully, as long as one is aware of its limitations as a linguistic guide.

[5] The *Nova Vulgata,* in conformity to the Nestle-Aland text, corrects this to "Spiritu Dei servimus."

[6] Witherington renders the verse as follows: *"For we are the circumcision, those who by the Spirit of God are worshiping and boasting in Christ Jesus and not being persuaded in the flesh."*

[7] Whether Χριστῷ Ἰησοῦ is understood to be the direct or indirect object of καυχώμενοι depends on how one interprets the preposition ἐν. As O'Brien notes, "the precise force of ἐν is disputed" (O'Brien 1991, p. 362). It is possible to interpret this as a "dative of sphere," similar to how πνεύματι θεοῦ is sometimes interpreted in the previous clause. Alternately, Χριστῷ Ἰησοῦ could also be understood to be the direct object καυχώμενοι, with ἐν being supplied because καυχάομαι demands it. See e.g., Jeremiah 9:21 (LXX), 1 Corinthians 1:31, 2 Corinthians 10:17, 2 Thessalonians 1:4, and James 1:9, where ἐν is consistently used before the direct object of καυχάομαι. O'Brien thus concludes in light of the aforementioned uses that Χριστῷ Ἰησοῦ is the object of boasting (O'Brien 1991, p. 362).

[8] For instances of καυχάομαι taking dative objects, see e.g., 1 Corinthians 1:31, 5:6, 2 Thessalonians 1:4, James 1:9, 4:16, etc.

[9] The question of whether we can speak of "deponency" in Koine Greek is a live one, with some scholars arguing that we should not speak of deponent verbs in NT Greek. Nevertheless, I still find it useful to use the term to refer to Greek verbs that are parsed as middle/passive mood, yet when translated into English yield an active meaning. For a discussion of this topic, see (Campbell 2015, pp. 91–104).

[10] Instructive for our purposes is the parallel use of the noun form of πείθω in 2 Corinthians 3:4, where Paul sets God through Christ as the proper object of one's confidence (Πεποίθησιν … ἔχομεν διὰ τοῦ Χριστοῦ πρὸς τὸν θεόν).

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
