# Peer review of "Christ Jesus as Object of Cultic Worship in Philippians 3:3b: A Linguistic Study"

_religions, doi:10.3390/rel16091100_

Round 1
Reviewer 1 Report
Comments and Suggestions for Authors
The article presents an interesting and valid research thesis on the worship of Christ in Phil 3:3. The author’s methodology is clearly defined and based on a linguistic and grammatical approach to the analyzed text. The paper has a well-organized structure. In the first paragraph, the author describes the linguistic approach, and in the subsequent paragraphs he/she analyses the context of Phil 3:3, the semantic domain of latreuo, and the object of latreuo in Phil 3:3. The author makes a plausible thesis that the object of the verb under study is Jesus Christ, which is important for the understanding of Paul’s Christology. I sympathize with the author’s position, but I believe that the paper and the argumentation need refinement in the following points:
1) The language is generally correct, but needs some revisions and stylistic corrections. See, for example, “Jobes posit (line 354)”. “I still find it useful to use (note 61)”.
2) The author essentially relies on the linguistics approach, which also includes discourse analysis and the verbal aspect of koine. While the paper contains the philological-grammatical analyses of the term latreuo, there is no reference to the Porter’s “pyramid” and placing Phil 3:3 in the broader context of Pauline discourse in the letter. Paragraph 3 of the paper is far too little. The author touches upon the issue of Paul’s Jewish opponents, but says nothing about the Philippians’ Christology and the function of Phil 3:3 in the chapter under study. What is the purpose of Christ’s worship in this part of Paul’s argument? What verbal aspect is implied in Paul’s use of latreuo? What does Paul want to express by using this term in the participial form, in pair with kauchaomai? How do these two verbs interact with each other, describing the worship given to Christ? The author’s comments in this regard need more depth and more bibliographical references.
3) The paper’s bibliography needs a thorough elaboration and expansion. While the author cites dictionaries and grammars, the article lacks references to many relevant commentaries, monographs, and articles on Philippians. One finds generic conclusions not followed by any footnotes or references to commentators, like: “Furthermore, of all the options that are being put forward, this option has the greatest number of commentators and scholars in its favor. (line 474)”. The same is true when discussing interpretive options in paragraph 4 – they should be better documented. I recommend consulting, e.g.:
Black, D. A., ‘The Discourse Structure of Philippians: A Study in Textlinguistics’, NovT 37 (1995) 16–49.
Blois, I. D., Mutual Boasting in Philippians: The Ethical Function of Shared Honor in Its Scriptural and Greco-Roman Context (LNTS 627; London/New York: T&T Clark, 2020).
Bockmuehl, M., The Epistle to the Philippians (BNTC; London: Continuum, 1997).
Cohick, L. H., ‘Philippians and Empire: Paul’s Engagement with Imperialism and the Imperial Cult’, Jesus Is Lord, Caesar Is Not: Evaluating Empire in New Testament Studies (ed. S. McKnight and J. B. Modica; Downers Grove, IL: InterVarsity, 2013) 167–83.
Davis, C. W., Oral Biblical Criticism: The Influence of the Principles of Orality on the Literary Structure of Paulʹs Epistle to the Philippians (JSNTSup 172; Sheffield: Sheffield Academic Press, 1999).
Fee, G. D., Paul’s Letter to the Philippians (NICNT; Grand Rapids, MI: Eerdmans, 1995).
Frey, J., Schliesser, B. and Niederhofer, V., ed., Der Philipperbrief des Paulus in der hellenistisch-römischen Welt WUNT (Tübingen: Mohr Siebeck, 2015).
Gupta, N. K., Worship that Makes Sense to Paul: A New Approach to the Theology and Ethics of Paul's Cultic Metaphors (BZNW 175; Berlin: Walter de Gruyter, 2nd ed., 20102).
Hawthorne, G. F. and Martin, R. P., Philippians: Revised and Expanded (WBC 43; Dallas: Word, 2004).
Martin, R. P., Carmen Christi: Philippians 2. 5-11 in Recent Interpretation and in the Setting of Early Christian Worship (SNTSMS 4; Cambridge: Cambridge University Press, 1967).
Oakes, P., ‘Re-Mapping the Universe: Paul and the Emperor in 1 Thessalonians and Philippians’, JSNT 27 (2005) 301–22.
O'Brien, P. T., The Epistle to the Philippians: A Commentary on the Greek Text (NIGTC; Grand Rapids, MI: Eerdmans, 1991).
Reumann, J., Philippians: A New Translation with Introduction and Commentary (AB 33B; New Haven/London: Yale University Press, 2008).
Silva, M., Philippians (BECNT; Grand Rapids, MI: Baker Academic, 2nd edn., 20052).
Witherington, B., Paul's Letter to the Philippians: A Socio-Rhetorical Commentary (Grand Rapids, MI: Eerdmans, 2011).
4) Finally, what does the reference to the worship of Christ in Phil 3:3 mean in the 1st century context? How does the worship of Jesus relate to the worship of emperors in Philippi? Why is Dunn opposing the cultic reference in Phil 3:3, and what arguments for it are given in Bauckham and Hurtado? How do they relate to the overall reading of Philippians? What is missing in the paper is the socio-religious context of Phil 3:3, which would explain why this issue is important and requires further research.
I congratulate the author on the idea and effort put into in the paper. Once the bibliography is expanded and the issues mentioned above are refined, the article will gain greatly in value and will be good for publication, contributing to the debate on Phil 3:3.
Author Response
- I have fixed the stylistic issues that have been mentioned.
- Unfortunately due to space limitations I could not say too much about this. I do briefly touch upon the issue of Paul's Christology in the conclusion, but mainly to draw out its implications for the broader debate upon early vs. late high Christology.
- I have updated the bibliography incorporating several of the commentaries listed. Though I did not cite all of them, I did find it particularly helpful to cite Bockmuehl, Fee, Hawthorne, O'Brien, and Witherington.
Reviewer 2 Report
Comments and Suggestions for Authors
This piece makes a bold argument, which is definitely worth considering. Some of the most compelling evidence is the application (via Wallace) of the Sharp’s Rule to the first two substantival participles in Phil 3:3, which pushes for a close association between these two activities. Those who worship are closely aligned with those who boast in, and both of these activities are enabled by the Spirit of God, and are centered around the person of Christ Jesus. The author needs to be more careful about how the boasting verb is discussed, particularly in relation to how the ἐν preposition functions grammatically with usage of the καυχάομαι verb. The author also must argue explicitly for why the negated substantival participle (πεποιθότες, “those who do not place confidence in...”) is separated off from the first two participles.

Author Response
I have responded to the main critique of the reviewer, which is the issue of whether Χριστῷ Ἰησοῦ functions as a direct objection for καυχάομαι in Phil. 3:3b. I have also incorporated an explicit discussion of πεποιθότες and why it is to be separated from the preceding participles. Finally, I fixed the typos mentioned at the beginning and incorporated several other of the minor revisions (such as removing the single citation of Varner).
Reviewer 3 Report
Comments and Suggestions for Authors
See attached.

Author Response
Both points brought up by the reviewer have been addressed. The quotation marks have been removed in the two points highlighted, and I significantly shortened the introductory material in pgs. 2-6.
Round 2
Reviewer 2 Report
Comments and Suggestions for Authors
Despite the added paragraph just prior to the conclusion, I still don’t think the author has provided adequate discussion of the rhetorical aspects that are changed when the 3 clause phrase is dropped down to being a 2 clause phrase. I also didn’t really see any discussion of pepoitho, which would be important for establishing the paper’s unique claim.
Also, here are a few typos that must be corrected:
Corrections Still Needed:
Line 38: “of the verb” should be “to the verb”
Line 43: “study [of]” latreuw
Line 293 and 286: twice in a row the paragraphs begin with “however”, which is poor style
Line 338: “for the when the”
Line 435: “by which worship [and boasting] take place”
Footnote 80: “direct object [of]” kauchomenoi, and also “1 Jeremiah, [1] Corinthians”
Line 519: “both verses [verbs]”
Author Response
I have fixed the typos that have been highlighted by the reviewer. I have also expanded the paragraph talking about the clause structure of 3:3b by noting how dividing the clause into two parts rather than three creates an antithetical parallelism in Paul's argument, as well as supplied a footnote pointing to 2 Corinthians 3:4 as a parallel usage of peithw.
Round 3
Reviewer 2 Report
Comments and Suggestions for Authors
The revisions seem to be adequate.